# Recurrent Ladder Networks

**Isabeau Prémont-Schwarz, Alexander Ilin, Tele Hotloo Hao,**
**Antti Rasmus, Rinu Boney, Harri Valpola**
The Curious AI Company
`{isabeau,alexilin,hotloo,antti,rinu,harri}@cai.fi`

## Abstract

We propose a recurrent extension of the Ladder networks [22] whose structure is motivated by the inference required in hierarchical latent variable models. We demonstrate that the recurrent Ladder is able to handle a wide variety of complex learning tasks that benefit from iterative inference and temporal modeling. The architecture shows close-to-optimal results on temporal modeling of video data, competitive results on music modeling, and improved perceptual grouping based on higher order abstractions, such as stochastic textures and motion cues. We present results for fully supervised, semi-supervised, and unsupervised tasks. The results suggest that the proposed architecture and principles are powerful tools for learning a hierarchy of abstractions, learning iterative inference and handling temporal information.

## 1 Introduction

Many cognitive tasks require learning useful representations on multiple abstraction levels. Hierarchical latent variable models are an appealing approach for learning a hierarchy of abstractions. The classical way of learning such models is by postulating an explicit parametric model for the distributions of random variables. The inference procedure, which evaluates the posterior distribution of the unknown variables, is then derived from the model – an approach adopted in probabilistic graphical models (see, e.g., [5]).

The success of deep learning can, however, be explained by the fact that popular deep models focus on learning the inference procedure directly. For example, a deep classifier like AlexNet [19] is trained to produce the posterior probability of the label for a given data sample. The representations that the network computes at different layers are related to the inference in an implicit latent variable model but the designer of the model does not need to know about them.

However, it is actually tremendously valuable to understand what kind of inference is required by different types of probabilistic models in order to design an efficient network architecture. Ladder networks [22, 28] are motivated by the inference required in a hierarchical latent variable model. By design, the Ladder networks aim to emulate a message passing algorithm, which includes a bottom-up pass (from input to label in classification tasks) and a top-down pass of information (from label to input). The results of the bottom-up and top-down computations are combined in a carefully selected manner.

The original Ladder network implements only one iteration of the inference algorithm but complex models are likely to require iterative inference. In this paper, we propose a recurrent extension of the Ladder network for iterative inference and show that the same architecture can be used for temporal modeling. We also show how to use the proposed architecture as an inference engine in more complex models which can handle multiple independent objects in the sensory input. Thus, the proposed architecture is suitable for the type of inference required by rich models: those that can learn a hierarchy of abstractions, can handle temporal information and can model multiple objects in the input.

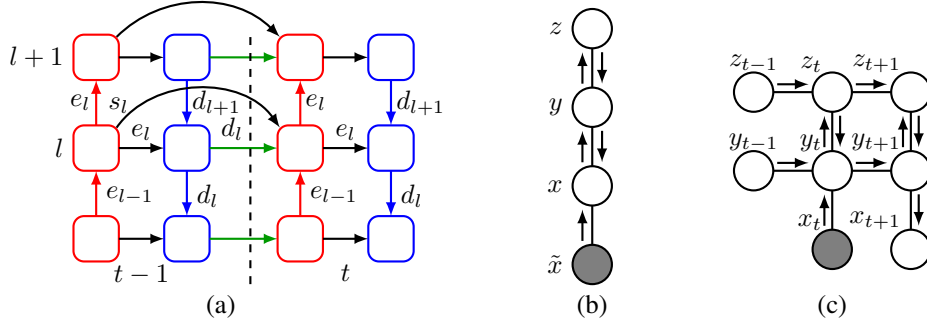

Figure 1: (a): The structure of the Recurrent Ladder networks. The encoder is shown in red, the decoder is shown in blue, the decoder-to-encoder connections are shown in green. The dashed line separates two iterations $t-1$ and $t$. (b)-(c): The type of hierarchical latent variable models for which RLadder is designed to emulate message passing. (b): A graph of a static model. (c): A fragment of a graph of a temporal model. White circles are unobserved latent variables, gray circles represent observed variables. The arrows represent the directions of message passing during inference.

## 2 Recurrent Ladder

**Recurrent Ladder networks**

In this paper, we present a recurrent extension of the Ladder networks which is conducive to iterative inference and temporal modeling. Recurrent Ladder (RLadder) is a recurrent neural network whose units resemble the structure of the original Ladder networks [22, 28] (see Fig. 1a). At every iteration $t$, the information first flows from the bottom (the input level) to the top through a stack of encoder cells. Then, the information flows back from the top to the bottom through a stack of decoder cells. Both the encoder and decoder cells also use the information that is propagated horizontally. Thus, at every iteration $t$, an encoder cell in the $l$-th layer receives three inputs: 1) the output of the encoder cell from the level below $e_{l-1}(t)$, 2) the output $d_l(t-1)$ of the decoder cell from the same level from the previous iteration, 3) the encoder state $s_l(t-1)$ from the same level from the previous iteration. It updates its state value $s_l(t)$ and passes the same output $e_l(t)$ both vertically and horizontally:

$$s_l(t) = f_{s,l}(e_{l-1}(t), d_l(t-1), s_l(t-1)) \tag{1}$$
$$e_l(t) = f_{e,l}(e_{l-1}(t), d_l(t-1), s_l(t-1)). \tag{2}$$

The encoder cell in the bottom layer typically sends observed data (possibly corrupted by noise) as its output $e_1(t)$. Each decoder cell is stateless, it receives two inputs (the output of the decoder cell from one level above and the output of the encoder cell from the same level) and produces one output

$$d_l(t) = g_l(e_l(t), d_{l+1}(t)), \tag{3}$$

which is passed both vertically and horizontally. The exact computations performed in the cells can be tuned depending on the task at hand. In practice, we have used LSTM [15] or GRU [8] cells in the encoder and cells inspired by the original Ladder networks in the decoder (see Appendix A).

Similarly to Ladder networks, the RLadder is usually trained with multiple tasks at different abstraction levels. Tasks at the highest abstraction level (like classification) are typically formulated at the highest layer. Conversely, the output of the decoder cell in the bottom level is used to formulate a low-level task which corresponds to abstractions close to the input. The low-level task can be denoising (reconstruction of a clean input from the corrupted one), other possibilities include object detection [21], segmentation [3, 23], or in a temporal setting, prediction. A weighted sum of the costs at different levels is optimized during training.

**Connection to hierarchical latent variables and message passing**

The RLadder architecture is designed to mimic the computational structure of an inference procedure in probabilistic hierarchical latent variable models. In an explicit probabilistic graphical model, inference can be done by an algorithm which propagates information (messages) between the nodes of a graphical model so as to compute the posterior distribution of the latent variables (see, e.g., [5]).

For static graphical models *implicitly* assumed by the RLadder (see Fig. 1b), messages need to be propagated from the input level up the hierarchy to the highest level and from the top to the bottom, as shown in Fig. 1a. In Appendix B, we present a derived iterative inference procedure for a simple static hierarchical model to give an example of a message-passing algorithm. We also show how that inference procedure can be implemented in the RLadder computational graph.

In the case of temporal modeling, the type of a graphical model assumed by the RLadder is shown in Fig. 1c. If the task is to do next step prediction of observations $x$, an online inference procedure should update the knowledge about the latent variables $y_t$, $z_t$ using observed data $x_t$ and compute the predictive distributions for the input $x_{t+1}$. Assuming that the distributions of the latent variables at previous time instances ($\tau < t$) are kept fixed, the inference can be done by propagating messages from the observed variables $x_t$ and the latent variables $y$, $z$ bottom-up, top-down and from the past to the future, as shown in Fig. 1c. The architecture of the RLadder (Fig. 1a) is designed so as to emulate such a message-passing procedure, that is the information can propagate in all the required directions: bottom-up, top-down and from the past to the future. In Appendix C, we present an example of the message-passing algorithm derived for a temporal hierarchical model to show how it is related to the RLadders's computation graph.

Even though the motivation of the RLadder architecture is to emulate a message-passing procedure, the nodes of the RLadder do not directly correspond to nodes of any specific graphical model.[1] The RLadder directly learns an inference procedure and the corresponding model is never formulated explicitly. Note also that using stateful encoder cells is not strictly motivated by the message-passing argument but in practice these skip connections facilitate training of a deep network.

As we mentioned previously, the RLadder is usually trained with multiple tasks formulated at different representation levels. The purpose of tasks is to encourage the RLadder to learn the right inference procedure, and hence formulating the right kind of tasks is crucial for the success of training. For example, the task of denoising encourages the network to learn important aspects of the data distribution [1, 2]. For temporal modeling, the task of next step prediction plays a similar role. The RLadder is most useful in problems that require accurate inference on multiple abstraction levels, which is supported by the experiments presented in this paper.

**Related work**

The RLadder architecture is similar to that of other recently proposed models for temporal modeling [10, 11, 9, 27, 20]. In [9], the recurrent connections (from time $t-1$ to time $t$) are placed in the lateral links between the encoder and the decoder. This can make it easier to extend an existing feed-forward network architecture to the case of temporal data as the recurrent units do not participate in the bottom-up computations. On the other hand, the recurrent units do not receive information from the top, which makes it impossible for higher layers to influence the dynamics of lower layers. The architectures in [10, 11, 27] are quite similar to ours but they could potentially derive further benefit from the decoder-to-encoder connections between successive time instances (green links in Fig. 1b). The aforementioned connections are well justified from the message-passing point of view: When updating the posterior distribution of a latent variable, one should combine the latest information from the top and from the bottom, and it is the decoder that contains the latest information from the top. We show empirical evidence to the importance of those connections in Section 3.1.

# 3  Experiments with temporal data

In this section, we demonstrate that the RLadder can learn an accurate inference algorithm in tasks that require temporal modeling. We consider datasets in which passing information both in time and in abstraction hierarchy is important for achieving good performance.

## 3.1  Occluded Moving MNIST

We use a dataset where we know how to do optimal inference in order to be able to compare the results of the RLadder to the optimal ones. To this end we designed the Occluded Moving MNIST

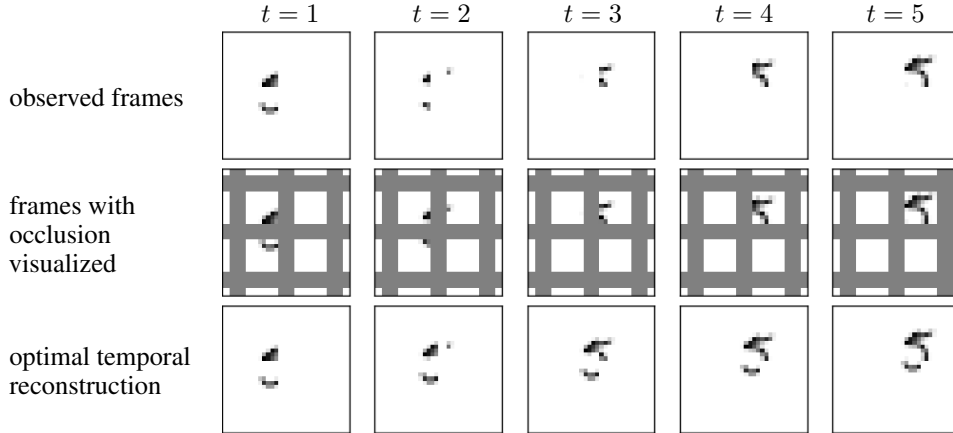

Figure 2: The Occluded Moving MNIST dataset. Bottom row: Optimal temporal recombination for a sequence of occluded frames from the dataset.

dataset. It consists of MNIST digits downscaled to $14 \times 14$ pixels flying on a $32 \times 32$ white background with white vertical and horizontal occlusion bars (4 pixels in width, and spaced by 8 visible pixels apart) which, when the digit flies behind them, occludes the pixels of the digit (see Fig. 2). We also restrict the velocities to be randomly chosen in the set of eight discrete velocities $\{(1, \pm 2), (-1, \pm 2), (2, \pm 1), (-2, \pm 1)\}$ pixels/frame, so that apart from the bouncing, the movement is deterministic. The digits are split into training, validation, and test sets according to the original MNIST split. The primary task is then to classify the digit which is only partially observable at any given moment, at the end of five time steps.

In order to do optimal classification, one would need to assimilate information about the digit identity (which is only partially visible at any given time instance) by keeping track of the observed pixels (see the bottom row of Fig. 2) and then feeding the resultant reconstruction to a classifier.

In order to encourage optimal inference, we add a next step prediction task to the RLadder at the bottom of the decoder: The RLadder is trained to predict the next *occluded* frame, that is the network never sees the un-occluded digit. This thus mimics a realistic scenario where the ground truth is not known. To assess the importance of the features of the RLadder, we also do an ablation study. In addition, we compare it to three other networks. In the first comparison network, the optimal reconstruction of the digit from the five frames (as shown in Fig. 2) is fed to a static feed-forward network from which the encoder of the RLadder was derived. This is our gold standard, and obtaining similar results to it implies doing close to optimal temporal inference. The second, a temporal baseline, is a deep feed-forward network (the one on which the encoder is based) with a recurrent neural network (RNN) at the top only so that, by design the network can propagate temporal information only at a high level, and not at a low level. The third, a hierarchical RNN, is a stack of convolutional LSTM units with a few convolutional layers in between, which is the RLadder amputated of its decoder. See Fig. 3 and Appendix D.1 for schematics and details of the architectures.

**Fully supervised learning results.** The results are presented in Table 1. The first thing to notice is that the RLadder reaches (up to uncertainty levels) the classification accuracy obtained by the network which was given the optimal reconstruction of the digit. Furthermore, if the RLadder does not have a decoder or the decoder-to-encoder connections, or if it is trained without the auxiliary prediction task, we see the classification error rise almost to the level of the temporal baseline. This means that even if a network has RNNs at the lowest levels (like with only the feed-forward encoder), or if it does not have a task which encourages it to develop a good world model (like the RLadder without the next-frame prediction task), or if the information cannot travel from the decoder to the encoder, the high level task cannot truly benefit from lower level temporal modeling.

Next, one notices from Table 1 that the top-level classification cost helps the low-level prediction cost in the RLadder (which in turn helps the top-level cost in a mutually beneficial cycle). This mutually supportive relationship between high-level and low-level inferences is nicely illustrated by the example in Fig. 4. Up until time step $t = 3$ inclusively, the network believes the digit to be a five

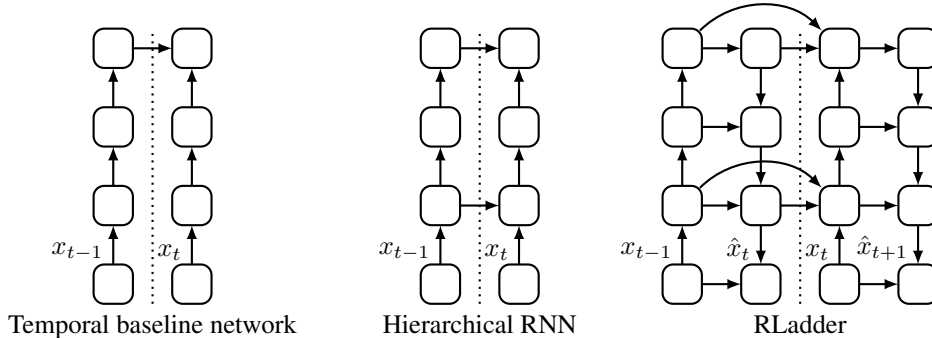

Figure 3: Architectures used for modeling occluded Moving MNIST. Temporal baseline network is a convolutional network with a fully connected RNN on top.

Table 1: Performance on Occluded Moving MNIST

|  | Classification error (%) | Prediction error, $\cdot 10^{-5}$ |
|---|---|---|
| Optimal reconstruction and static classifier | $0.71 \pm 0.03$ | |
| Temporal baseline | $2.02 \pm 0.16$ | |
| Hierarchical RNN (encoder only) | $1.60 \pm 0.05$ | |
| RLadder w/o prediction task | $1.51 \pm 0.21$ | |
| RLadder w/o decoder-to-encoder conn. | $1.24 \pm 0.05$ | $156.7 \pm 0.4$ |
| RLadder w/o classification task | | $155.2 \pm 2.5$ |
| RLadder | $\mathbf{0.74 \pm 0.09}$ | $\mathbf{150.1 \pm 0.1}$ |

(Fig. 4a). As such, at $t = 3$, the network predicts that the top right part of the five which has been occluded so far will stick out from behind the occlusions as the digit moves up and right at the next time step (Fig. 4b). Using the decoder-to-encoder connections, the decoder can relay this expectation to the encoder at $t = 4$. At $t = 4$ the encoder can compare this expectation with the actual input where the top right part of the five is absent (Fig. 4c). Without the decoder-to-encoder connections this comparison would have been impossible. Using the upward path of the encoder, the network can relay this discrepancy to the higher classification layers. These higher layers with a large receptive field can then conclude that since it is not a five, then it must be a three (Fig. 4d). Now thanks to the decoder, the higher classification layers can relay this information to the lower prediction layers so that they can change their prediction of what will be seen at $t = 5$ appropriately (Fig. 4e). Without a decoder which would bring this high level information back down to the low level, this drastic update of the prediction would be impossible. With this information the lower prediction layer can now predict that the top-left part of the three (which it has never seen before) will appear at the next time step from behind the occlusion, which is indeed what happens at $t = 5$ (Fig. 4f).

**Semi-supervised learning results.** In the following experiment, we test the RLadder in the semi-supervised scenario when the training data set contains 1.000 labeled sequences and 59.000 unlabeled ones. To make use of the unlabeled data, we added an extra auxiliary task at the top level which was the consistency cost with the targets provided by the Mean Teacher (MT) model [26]. Thus, the RLadder was trained with three tasks: 1) next step prediction at the bottom, 2) classification at the top, 3) consistency with the MT outputs at the top. As shown in Table 2, the RLadder improves dramatically by learning a better model with the help of unlabeled data independently and in addition to other semi-supervised learning methods. The temporal baseline model also improves the classification accuracy by using the consistency cost but it is clearly outperformed by the RLadder.

## 3.2 Polyphonic Music Dataset

In this section, we evaluate the RLadder on the midi dataset converted to piano rolls [6]. The dataset consists of piano rolls (the notes played at every time step, where a time step is, in this case, an eighth note) of various piano pieces. We train an 18-layer RLadder containing five convolutional LSTMs and one fully-connected LSTM. More details can be found in Appendix D.2. Table 3 shows the

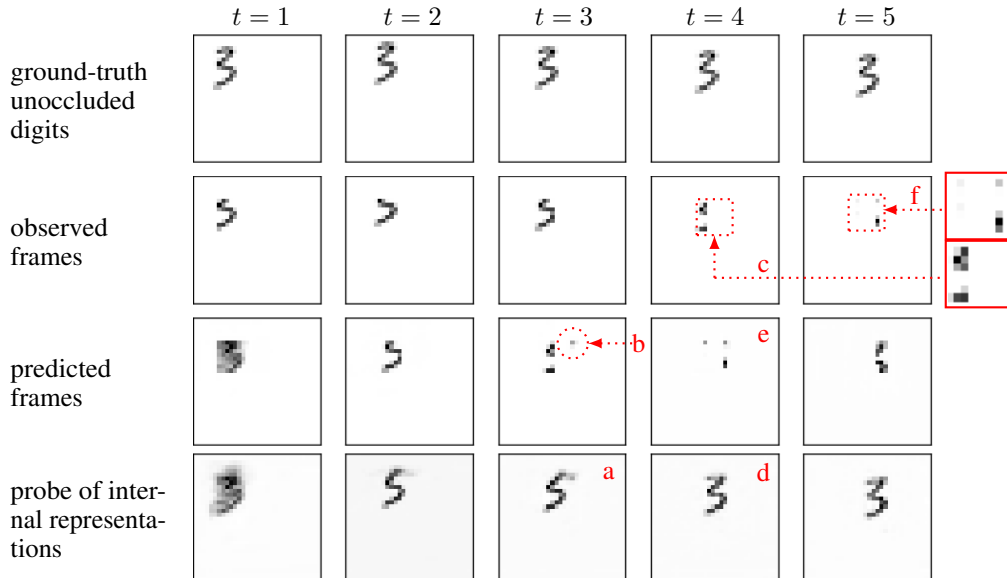

Figure 4: Example prediction of an RLadder on the occluded moving MNIST dataset. First row: the ground truth of the digit, which the network never sees and does not train on. Second row: The actual five frames seen by the network and on which it trains. Third row: the predicted next frames of a trained RLadder. Fourth row: A stopped-gradient (gradient does not flow into the RLadder) readout of the bottom layer of the decoder trained on the ground truth to probe what aspects of the digit are represented by the neurons which predict the next frame. Notice how at $t = 1$, the network does not yet know in which direction the digit will move and so it predicts a superposition of possible movements. Notice further (red annotations a-f), that until $t = 3$, the network thought the digit was a five, but when the top bar of the supposed five did not materialize on the other side of the occlusion as expected at $t = 4$, the network immediately concluded correctly that it was actually a three.

Table 2: Classification error (%) on semi-supervised Occluded Moving MNIST

| | 1k labeled | 1k labeled & 59k unlabeled w/o MT | MT |
|---|---|---|---|
| Optimal reconstruction and static classifier | $3.50 \pm 0.28$ | $3.50 \pm 0.28$ | $1.34 \pm 0.04$ |
| Temporal baseline | $10.86 \pm 0.43$ | $10.86 \pm 0.43$ | $3.14 \pm 0.16$ |
| RLadder | $10.49 \pm 0.81$ | $5.20 \pm 0.77$ | $1.69 \pm 0.14$ |

negative log-likelhoods of the next-step prediction obtained on the music dataset, where our results are reported as mean plus or minus standard deviation over 10 seeds. We see that the RLadder is competitive with the best results, and gives the best results amongst models outputting the marginal distribution of notes at each time step.

The fact that the RLadder did not beat [16] on the midi datasets shows one of the limitations of RLadder. Most of the models in Table 3 output a joint probability distribution of notes, unlike RLadder which outputs the marginal probability for each note. That is to say, those models, to output the probability of a note, take as input the notes at previous time instances, but also the ground truth of the notes to the left at the same time instance. RLadder does not do that, it only takes as input the past notes played. Even though, as the example in 3.1 of the the digit five turning into a three after seeing only one absent dot, shows that internally the RLadder models the joint distribution.

## 4 Experiments with perceptual grouping

In this section, we show that the RLadder can be used as an inference engine in a complex model which benefits from iterative inference and temporal modeling. We consider the task of perceptual grouping, that is identifying which parts of the sensory input belong to the same higher-level perceptual

Table 3: Negative log-likelihood (smaller is better) on polyphonic music dataset

|  | Piano-midi.de | Nottingham | Muse | JSB Chorales |
|---|---|---|---|---|
| **Models outputting a joint distribution of notes:** | | | | |
| NADE masked [4] | 7.42 | 3.32 | 6.48 | 8.51 |
| NADE [4] | 7.05 | 2.89 | 5.54 | 7.59 |
| RNN-RBM [6] | 7.09 | 2.39 | 6.01 | 6.27 |
| RNN-NADE (HF) [6] | 7.05 | 2.31 | 5.60 | **5.56** |
| LSTM-NADE [16] | 7.39 | 2.06 | 5.03 | 6.10 |
| TP-LSTM-NADE [16] | 5.49 | 1.64 | 4.34 | 5.92 |
| BALSTM [16] | **5.00** | **1.62** | **3.90** | 5.86 |
| **Models outputting marginal probabilities for each note:** | | | | |
| RNN [4] | 7.88 | 3.87 | 7.43 | 8.76 |
| LSTM [17] | 6.866 | 3.492 | | |
| MUT1 [17] | 6.792 | 3.254 | | |
| RLadder | **6.19 ± 0.02** | **2.42 ± 0.03** | **5.69 ± 0.02** | **5.64 ± 0.02** |

components (objects). We enhance the previously developed model for perceptual grouping called Tagger [13] by replacing the originally used Ladder engine with the RLadder. For another perspective on the problem see [14] which also extends Tagger to a recurrent neural network, but does so from an expectation maximization point of view.

## 4.1 Recurrent Tagger

Tagger is a model designed for perceptual grouping. When applied to images, the modeling assumption is that each pixel $\tilde{x}_i$ belongs to one of the $K$ objects, which is described by binary variables $z_{i,k}$: $z_{i,k} = 1$ if pixel $i$ belongs to object $k$ and $z_{i,k} = 0$ otherwise. The reconstruction of the whole image using object $k$ only is $\boldsymbol{\mu}_k$ which is a vector with as many elements $\mu_{i,k}$ as there are pixels. Thus, the assumed probabilistic model can be written as follows:

$$p(\tilde{\mathbf{x}}, \boldsymbol{\mu}, \mathbf{z}, \mathbf{h}) = \prod_{i,k} \mathcal{N}(\tilde{x}_i | \mu_{i,k}, \sigma_k^2)^{z_{i,k}} \prod_{k=1}^{K} p(\mathbf{z}_k, \boldsymbol{\mu}_k | \mathbf{h}_k) p(\mathbf{h}_k) \tag{4}$$

where $\mathbf{z}_k$ is a vector of elements $z_{i,k}$ and $\mathbf{h}_k$ is (a hierarchy of) latent variables which define the shape and the texture of the objects. See Fig. 5a for a graphical representation of the model and Fig. 5b for possible values of model variables for the textured MNIST dataset used in the experiments of Section 4.2. The model in (4) is defined for *noisy* image $\tilde{\mathbf{x}}$ because Tagger is trained with an auxiliary low-level task of denoising. The inference procedure in model (4) should evaluate the posterior distributions of the latent variables $\mathbf{z}_k$, $\boldsymbol{\mu}_k$, $\mathbf{h}_k$ for each of the $K$ groups given corrupted data $\tilde{\mathbf{x}}$. Making the approximation that the variables of each of the $K$ groups are independent a posteriori

$$p(\mathbf{z}, \boldsymbol{\mu}, \mathbf{h} | \tilde{\mathbf{x}}) \approx \prod_k q(\mathbf{z}_k, \boldsymbol{\mu}_k, \mathbf{h}_k), \tag{5}$$

the inference procedure could be implemented by iteratively updating each of the $K$ approximate distributions $q(\mathbf{z}_k, \boldsymbol{\mu}_k, \mathbf{h}_k)$, if the model (4) and the approximation (5) were defined explicitly.

Tagger does not explicitly define a probabilistic model (4) but learns the inference procedure directly. The iterative inference procedure is implemented by a computational graph with $K$ copies of the same Ladder network doing inference for one of the groups (see Fig. 5c). At the end of every iteration, the inference procedure produces the posterior probabilities $\pi_{i,k}$ that pixel $i$ belongs to object $k$ and the point estimates of the reconstructions $\boldsymbol{\mu}_k$ (see Fig. 5c). Those outputs, are used to form the low-level cost and the inputs for the next iteration (see more details in [13]). In this paper, we replace the original Ladder engine of Tagger with the RLadder. We refer to the new model as RTagger.

## 4.2 Experiments on grouping using texture information

The goal of the following experiment is to test the efficiency of RTagger in grouping objects using the texture information. To this end, we created a dataset that contains thickened MNIST digits with

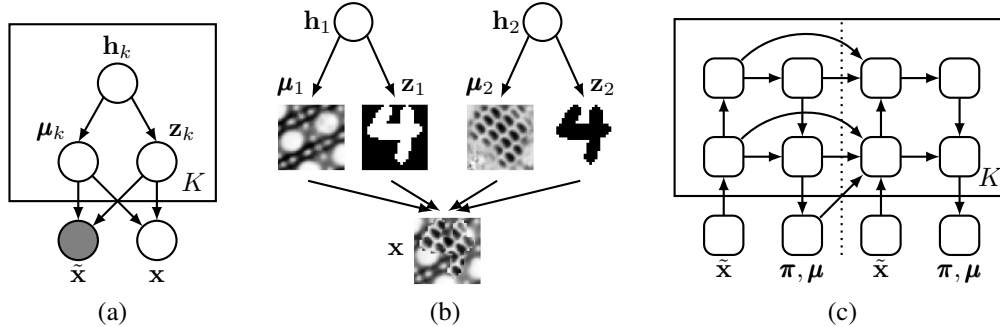

(a)    (b)    (c)

Figure 5: (a): Graphical model for perceptual grouping. White circles are unobserved latent variables, gray circles represent observed variables. (b): Examples of possible values of model variables for the textured MNIST dataset. (c): Computational graph that implements iterative inference in perceptual grouping task (RTagger). Two graph iterations are drawn. The plate notation represent $K$ copies of the same graph.

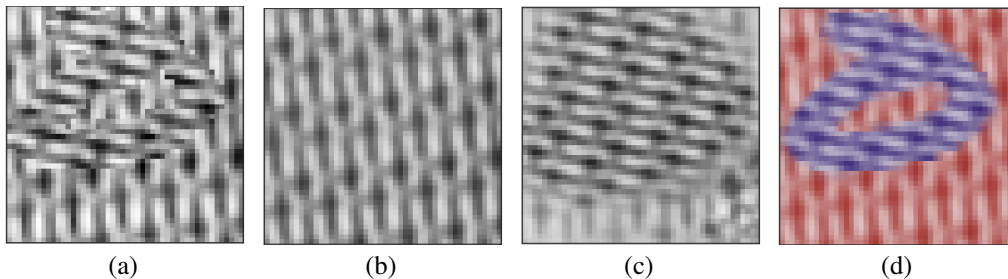

(a)    (b)    (c)    (d)

Figure 6: (a): Example image from the Brodatz-textured MNIST dataset. (b): The image reconstruction $\mathbf{m}_0$ by the group that learned the background. (c): The image reeconstruction $\mathbf{m}_1$ by the group that learned the digit. (d): The original image colored using the found grouping $\boldsymbol{\pi}_k$.

20 textures from the Brodatz dataset [7]. An example of a generated image is shown in Fig. 6a. To create a greater diversity of textures (to avoid over-fitting), we randomly rotated and scaled the 20 Brodatz textures when producing the training data.

The network trained on the textured MNIST dataset has the architecture presented in Fig. 5c with three iterations. The number of groups was set to $K = 3$. The details of the RLadder architecture are presented in Appendix D.3. The network was trained on two tasks: The low-level segmentation task was formulated around denoising, the same way as in the Tagger model [13]. The top-level cost was the log-likelihood of the digit class at the last iteration.

Table 4 presents the obtained performance on the textured MNIST dataset in both fully supervised and semi-supervised settings. All experiments were run over 5 seeds. We report our results as mean plus or minus standard deviation. In some runs, Tagger experiments did not converge to a reasonable solution (because of unstable or too slow convergence), so we did not include those runs in our evaluations. Following [13], the segmentation accuracy was computed using the adjusted mutual information (AMI) score [29] which is the mutual information between the ground truth segmentation and the estimated segmentation $\boldsymbol{\pi}_k$ scaled to give one when the segmentations are identical and zero when the output segmentation is random.

For comparison, we trained the Tagger model [13] on the same dataset. The other comparison method was a feed-forward convolutional network which had an architecture resembling the bottom-up pass (encoder) of the RLadder and which was trained on the classification task only. One thing to notice is that the results obtained with the RTagger clearly improve over iterations, which supports the idea that iterative inference is useful in complex cognitive tasks. We also observe that RTagger outperforms Tagger and both approaches significantly outperform the convolutional network baseline in which the classification task is not supported by the input-level task. We have also observed that the top-level classification tasks makes the RTagger faster to train in terms of the number of updates, which also supports that the high-level and low-level tasks mutually benefit from each other: Detecting object

Table 4: Results on the Brodatz-textured MNIST. $i$-th column corresponds to the intermediate results of RTagger after the $i$-th iteration. In the fully supervised case, Tagger was only trained successfully in 2 of the 5 seeds, the given results are for those 2 seeds. In the semi-supervised case, we were not able to train Tagger successfully.

| 50k labeled | | |
| --- | --- | --- |
| **Segmentation accuracy, AMI:** | | |
| RTagger | 0.55 | 0.75 | **0.80 ± 0.01** |
| Tagger | – | – | 0.73 ± 0.02 |
| **Classification error, %:** | | |
| RTagger | 18.2 | 8.0 | **5.9 ± 0.2** |
| Tagger | – | – | 12.15 ± 0.1 |
| ConvNet | – | – | 14.3 ± 0.46 |

| 1k labeled + 49k unlabeled | | |
| --- | --- | --- |
| **Segmentation accuracy, AMI:** | | |
| RTagger | 0.56 | 0.74 | **0.80 ± 0.03** |
| **Classification error, %:** | | |
| RTagger | 63.8 | 28.2 | **22.6 ± 6.2** |
| ConvNet | – | – | 88 ± 0.30 |

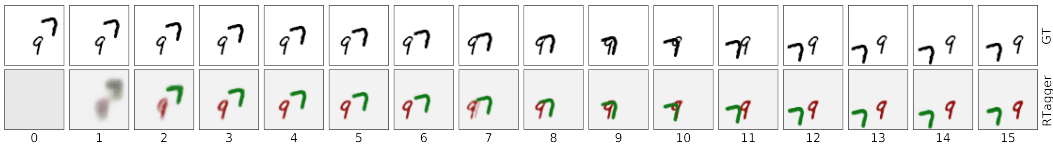

Figure 7: Example of segmentation and generation by the RTagger trained on the Moving MNIST. First row: frames 0-9 is the input sequence, frames 10-15 is the ground truth future. Second row: Next step prediction of frames 1-9 and future frame generation (frames 10-15) by RTagger, the colors represent grouping performed by RTagger.

boundaries using textures helps classify a digit, while knowing the class of the digit helps detect the object boundaries. Figs. 6b-d show the reconstructed textures and the segmentation results for the image from Fig. 6a.

## 4.3 Experiments on grouping using movement information

The same RTagger model can perform perceptual grouping in video sequences using motion cues. To demonstrate this, we applied the RTagger to the moving MNIST [25][2] sequences of length 20 and the low-level task was prediction of the next frame. When applied to temporal data, the RTagger assumes the existence of $K$ objects whose dynamics are independent of each other. Using this assumption, the RTagger can separate the two moving digits into different groups. We assessed the segmentation quality by the AMI score which was computed similarly to [13, 12] ignoring the background in the case of a uniform zero-valued background and overlap regions where different objects have the same color. The achieved averageAMI score was 0.75. An example of segmentation is shown in Fig. 7. When we tried to use Tagger on the same dataset, we were only able to train successfully in a single seed out of three. This is possibly because speed is intermediate level of abstraction not represented at the pixel level. Due to its reccurent connections, RTagger can keep those representations from one time step to the next and segment accordingly, something more difficult for Tagger to do, which might explain the training instability.

## 5 Conclusions

In the paper, we presented recurrent Ladder networks. The proposed architecture is motivated by the computations required in a hierarchical latent variable model. We empirically validated that the recurrent Ladder is able to learn accurate inference in challenging tasks which require modeling dependencies on multiple abstraction levels, iterative inference and temporal modeling. The proposed model outperformed strong baseline methods on two challenging classification tasks. It also produced competitive results on a temporal music dataset. We envision that the purposed Recurrent Ladder will be a powerful building block for solving difficult cognitive tasks.

**Acknowledgments**

We would like to thank Klaus Greff and our colleagues from The Curious AI Company for their contribution in the presented work, especially Vikram Kamath and Matti Herranen.

## Footnotes

[1]To emphasize this, we used different shapes for the nodes of the RLadder network (Fig. 1a) and the nodes of graphical models that inspired the RLadder architecture (Figs. 1b-c).

[2]For this experiment, in order to have the ground truth segmentation, we reimplemented the dataset ourselves.

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
