[Supplementary Material]

## A Cells used in the decoder of the RLadder

Decoder cells receive two inputs: vector $\mathbf{v}_{\text{top}}$ from the top and vector $\mathbf{h}$ from the encoder and produce output $\mathbf{y}$.

### A.1 G1 and convG1 cells

$$\mathbf{u} = \text{BN}(\mathbf{A}\mathbf{v}_{\text{top}} + \mathbf{b})$$
$$\mathbf{s} = f(\mathbf{u}, \mathbf{w}_s)$$
$$\mathbf{y} = \mathbf{s} \odot \mathbf{h} + (1 - \mathbf{s}) \odot f(\mathbf{u}, \mathbf{w}),$$

where $\text{BN}()$ is batch normalization and

$$f(\mathbf{u}, \mathbf{w}) = w_0 \, \text{sigmoid}(w_1 \mathbf{u} + w_2) + w_3 \mathbf{u} + w_4 \tag{6}$$

with $w_i$ being the elements of vector $\mathbf{w}$ and $\odot$ the element-wise product. In the convolutional version ConvG1 of that cell, $\mathbf{u}$ in the first formula is computed with the convolution operation $*$:

$$\mathbf{u} = \mathbf{A} * \mathbf{v}_{\text{top}} + \mathbf{b}.$$

### A.2 convG2 cell

$$\boldsymbol{\mu}_1 = f(\text{LN}(\mathbf{A}_1 * \mathbf{v}_{\text{top}}) + \text{LN}(\mathbf{B}_1 * \mathbf{h}) + \mathbf{c}_1, \mathbf{w}_1)$$
$$\boldsymbol{\mu}_2 = f(\text{LN}(\mathbf{A}_2 * \mathbf{v}_{\text{top}}) + \text{LN}(\mathbf{B}_2 * \mathbf{h}) + \mathbf{c}_2, \mathbf{w}_2)$$
$$\mathbf{s} = f(\text{LN}(\mathbf{A}_s * \mathbf{v}_{\text{top}}) + \text{LN}(\mathbf{B}_s * \mathbf{h}) + \mathbf{c}_s, \mathbf{w}_s)$$
$$\mathbf{y} = \mathbf{s} \odot \boldsymbol{\mu}_1 + (1 - \mathbf{s}) \odot \boldsymbol{\mu}_2$$

where $f$ is defined in (6) and $\text{LN}()$ is layer normalization.

### A.3 convG3 cell

$$\mathbf{u} = \text{relu}(\text{LN}(\mathbf{A} * \mathbf{v}_{\text{top}}) + \text{LN}(\mathbf{B} * \mathbf{h}) + \mathbf{c})$$
$$\mathbf{s} = \text{sigmoid}(\mathbf{W}_s * \mathbf{u})$$
$$\mathbf{y} = \mathbf{s} \odot (\mathbf{D} * \mathbf{u}) + (1 - \mathbf{s}) \odot (\mathbf{E} * \mathbf{u}),$$

where $\text{LN}()$ is layer normalization.

### A.4 Detailes of normalizations used

**Encoder:** For all the experiments all non-recurrent layers in the encoder, whether convolutional or not, are normalized using batch normalization. In the convolution LSTM, there is a layer normalization after the all the first convolutions of the inputs in the RLadder experiments and batch normalization in the RTagger experiments, these normalizations were important to get things working.

**Decoder:** In the decoder gating functions, there is always a layer normalization after the first convolution over the concatenation of the inputs in the RLadder experiments, and this normalization was replaced by batch normalization in the RTagger experiments. A normalization at this point was critical for optimal performance.

## B Example of approximate inference in a static model

The structure of the RLadder is designed to emulate a message-passing algorithm in a hierarchical latent variable model. To illustrate this, let us consider a simple hierarchical latent variable model with three one-dimensional variables whose joint probability distribution is given by

$$p(x, y, z) = p(z)p(y|z)p(x|y) \tag{7}$$
$$p(x|y) = \mathcal{N}(x|w_x y, \sigma_x^2) \tag{8}$$
$$p(y|z) = \mathcal{N}(y|w_{yz} z, \sigma_y^2) \tag{9}$$
$$p(z) = \mathcal{N}(z|\mu_z, \sigma_z^2), \tag{10}$$

Figure 8: (a): Simple hierarchical latent variable model. A filled node represents an observed variable. (b): Directions of message propagation in an inference procedure. (c): Computational graph which implements an iterative inference procedure has the RLadder architecture.

where $\mathcal{N}(\cdot|m, v)$ denotes the Gaussian probability density function with mean $m$ and variance $v$. We want to derive a denoising algorithm that recovers clean observation $x$ from its corrupted version $\tilde{x}$, where the corruption is also modeled to be Gaussian:

$$p(\tilde{x}|x) = \mathcal{N}(\tilde{x}|x, \sigma^2). \tag{11}$$

The reason we look at denoising is because denoising is a task which can be used for unsupervised learning of the data distribution [1, 2]. The graphical representation of the model shown in Fig. 8a.

In order to do optimal denoising, one needs to evaluate the expectation of $p(x|\tilde{x})$, which can be done by learning the joint posterior distribution of the unknown variables $x$, $y$ and $z$. For this linear Gaussian model, it is possible to derive an inference algorithm that is guaranteed to produce the exact posterior distribution in a finite number of steps (see, e.g., [5]). However, in more complex models, the posterior distribution of the latent variables is impossible to represent exactly and therefore a neural network that learns the inference procedure needs to represent an approximate distribution. To this end, we derive an *approximate* inference procedure for this simple probabilistic model.

Using the variational Bayesian approach, we can approximate the joint posterior distribution by a distribution of a simpler form. For example, all the latent variables can be modeled to be independent Gaussian variables a posteriori:

$$p(x, y, z|\tilde{x}) \approx q(x, y, z) = q(x)q(y)q(z), \tag{12}$$
$$q(x) = \mathcal{N}(x|m_x, v_x), \tag{13}$$
$$q(y) = \mathcal{N}(y|m_y, v_y), \tag{14}$$
$$q(z) = \mathcal{N}(z|m_z, v_z). \tag{15}$$

and the goal of the inference procedure is to estimate parameters $m_x$, $v_x$, $m_y$, $v_y$, $m_z$, $v_z$ of the approximate posterior (12)–(15) for the model in (7)–(11) with fixed parameters $w_x$, $\sigma_x^2$, $w_{yz}$, $\sigma_y^2$, $\mu_z$, $\sigma_z^2$, $\sigma^2$.

The optimal posterior approximation $q(x, y, z)$ can be found by minimizing the lower bound of the Kullback-Leibler divergence between the true distribution and the approximation:

$$C_{\text{VB}} = \int \log \frac{q(x)q(y)q(z)}{p(x, y, z, \tilde{x})} q(x)q(y)q(z) dy dz = \langle \log q(x)q(y)q(z) \rangle - \langle \log p(x, y, z, \tilde{x}) \rangle \tag{16}$$

where $\langle \cdot \rangle$ denotes the expectation over $q(x, y, z)$. This can be done with the following iterative procedure:

$$m_x = s_x \tilde{x} + (1 - s_x) w_y m_y \qquad s_x = \text{sigmoid}(\log(\sigma_x^2/\sigma^2)) \qquad v_x^{-1} = \frac{1}{\sigma^2} + \frac{1}{\sigma_x^2} \tag{17}$$

$$m_y = s_y \frac{m_x}{w_x} + (1 - s_y) w_{yz} m_z \qquad s_y = \text{sigmoid}(\log(\sigma_y^2 w_x^2/\sigma_x^2)) \qquad v_y^{-1} = \frac{w_x^2}{\sigma_x^2} + \frac{1}{\sigma_y^2} \tag{18}$$

$$m_z = s_z \frac{m_y}{w_{yz}} + (1 - s_z) \mu_z \qquad s_z = \text{sigmoid}(\log(\sigma_z^2 w_{yz}^2/\sigma_y^2)) \qquad v_z^{-1} = \frac{w_{yz}^2}{\sigma_y^2} + \frac{1}{\sigma_z^2}. \tag{19}$$

Figure 9: (a): Fragment of a graph of a temporal model relevant for updating the distributions of unknown variables after observing $x_t$. Light-gray circles represent latent variables whose distribution is not updated. (b): Directions of information propagation needed for inference. (c): The structure of the RLadder network can be seen as a computational graph implementing information flow in (b). The dotted arrows are the skip connections that would be needed if we forced the activations to be literally interpreted as the distribution parameters of the latent variables.

Thus, in order to update the posterior distribution of a latent variable, one needs to combine the information coming from one level above and from one level below. For example, in order to update $q(y)$, one needs information from below ($m_x/w_x$ and $w_x^2/\sigma_x^2$) and from above ($w_{yz}m_z$ and $1/\sigma_y^2$). We can think of the information needed for updating the parameters describing the posterior distributions of the latent variables as 'messages' propagating between the nodes of a probabilistic graphical model (see Fig. 8b). Since there are mutual dependencies between $m_x$, $m_y$ and $m_z$ in (17)–(19), the update rules need to be iterated multiple times until convergence. This procedure can be implemented using the RLadder computational graph with the messages shown in Fig. 8c. Note that in practice, the computations used in the cells of the RLadder are not dictated by any particular explicit probabilistic model. The original Ladder networks [22] contain only one iteration of the bottom-up and top-down passes. The RLadder extends the model to multiple passes. Note also that the computations used in the decoder (top-down pass) of the Ladder networks are inspired by the gating structure of the update rules (17)–(19) in simple Gaussian models.

## C   Example of approximate inference in a simple temporal model

In this section we consider a simple hierarchical and temporal model and look at the relationship between temporal inferance in that model and the computational structure of RLadder. Consider a simple probabilistic model with three levels of hierarchy $x_t$, $y_t$, $z_t$ in which variables vary in time (see Fig. 9a). The conditional distributions given the latent variables in the past $y_{t-1}$, $z_{t-1}$ are defined as follows:

$$p(x_t|y_t) = \mathcal{N}(x_t|w_x x_{t-1} + w_{xy} y_t, \sigma_x^2) \tag{20}$$

$$p(y_t|y_{t-1}, z_t) = \mathcal{N}(y_t|w_y y_{t-1} + w_{yz} z_t, \sigma_y^2) \tag{21}$$

$$p(z_t|z_{t-1}) = \mathcal{N}(z_t|w_z z_{t-1}, \sigma_z^2) \tag{22}$$

At time $t-1$ the observed variables are $x_{1..t-1} = (x_1, ..., x_{t-1})$. Using the variational Bayesian approach, we can approximate the joint posterior distribution of the latent variables $y$ and $z$ by a distribution of a simpler form:

$$p(y_{t-1}, z_{t-1}|x_{1..t-1}) \approx q(y_{t-1})q(z_{t-1})$$
$$q(y_{t-1}) = \mathcal{N}(y_{t-1}|m_{y,t-1}, v_{y,t-1})$$
$$q(z_{t-1}) = \mathcal{N}(z_{t-1}|m_{z,t-1}, v_{z,t-1}).$$

The cost function minimized in the variational Bayesian approach is:

$$C_{\text{VB}} = \langle \log q(y_t)q(z_t) - \log p(x_t|y_t)p(y_t|y_{t-1}, z_t)p(z_t|z_{t-1}) - \log p(y_{t-1})p(z_{t-1}) \rangle$$

where the expectation is taken over $q(y_t, z_t, y_{t-1}, z_{t-1}, \ldots, y_1, z_1) = \prod_{s=1}^{t} q(y_s)q(z_s)$. Let us elaborate the terms of $C_{\text{VB}}$. The first term is

$$
\begin{aligned}
C_q &= \left\langle -\frac{1}{2}\log 2\pi v_{y,t} - \frac{1}{2v_{y,t}}(y_t - m_{y,t})^2 - \frac{1}{2}\log 2\pi v_{z,t} - \frac{1}{2v_{z,t}}(z_t - m_{z,t})^2 \right\rangle \\
&= -\frac{1}{2}\log 2\pi v_{y,t} - \frac{1}{2} - \frac{1}{2}\log 2\pi v_{z,t} - \frac{1}{2}.
\end{aligned}
$$

The second term is

$$
\begin{aligned}
C_p &= - \left\langle -\frac{1}{2}\log 2\pi\sigma_x^2 - \frac{1}{2\sigma_x^2}(x_t - w_x y_t)^2 - \frac{1}{2}\log 2\pi\sigma_y^2 - \frac{1}{2\sigma_y^2}(y_t - w_y y_{t-1} - w_{yz} z_t)^2 \right\rangle \\
&\quad - \left\langle -\frac{1}{2}\log 2\pi\sigma_z^2 - \frac{1}{2\sigma_z^2}(z_t - w_z z_{t-1})^2 \right\rangle \\
&= \frac{1}{2}\log 2\pi\sigma_x^2 + \frac{1}{2\sigma_x^2}(x_t^2 - 2x_t w_x m_{y,t} + w_x^2(m_{y,t}^2 + v_{y,t})) \\
&\quad + \frac{1}{2}\log 2\pi\sigma_y^2 + \frac{1}{2\sigma_y^2}(m_{y,t}^2 + v_{y,t} + w_y^2(m_{y,t-1}^2 + v_{y,t-1}) + w_{yz}^2(m_{z,t}^2 + v_{z,t}) \\
&\quad - 2m_{y,t}w_y m_{y,t-1} - 2m_{y,t}w_{yz}m_{z,t} + 2w_y m_{y,t-1}w_{yz}m_{z,t}) \\
&\quad + \frac{1}{2}\log 2\pi\sigma_z^2 + \frac{1}{2\sigma_z^2}(m_{z,t}^2 + v_{z,t} - 2m_{z,t}w_z m_{z,t-1} + w_z^2(m_{z,t-1}^2 + v_{z,t-1})).
\end{aligned}
$$

The last term is a function of variational parameters $m_{y,t-1}, m_{z,t-1}, v_{y,t-1}, v_{z,t-1}$ which we do not update at time instance $t$. Taking the derivative of $C_{\text{VB}}$ wrt the variational parameters yields:

$$
\begin{aligned}
\frac{\partial C_{\text{VB}}}{\partial m_{y,t}} &= \frac{1}{\sigma_x^2}(-x_t w_x + w_x^2 m_{y,t}) + \frac{1}{\sigma_y^2}(m_{y,t} - w_y m_{y,t-1} - w_{yz} m_{z,t}) \\
\frac{\partial C_{\text{VB}}}{\partial v_{y,t}} &= -\frac{1}{2v_{y,t}} + \frac{w_x^2}{2\sigma_x^2} + \frac{1}{2\sigma_y^2} \\
\frac{\partial C_{\text{VB}}}{\partial m_{z,t}} &= \frac{w_{yz}}{\sigma_y^2}(w_{yz} m_{z,t} - m_{y,t} + w_y m_{y,t-1}) + \frac{1}{\sigma_z^2}(m_{z,t} - w_z m_{z,t-1}) \\
\frac{\partial C_{\text{VB}}}{\partial v_{z,t}} &= -\frac{1}{2v_z} + \frac{w_{yz}^2}{2\sigma_y^2} + \frac{1}{2\sigma_z^2}.
\end{aligned}
$$

Equating the derivatives to zero yields:

$$
\begin{aligned}
v_{y,t} &= \left( \frac{w_x^2}{\sigma_x^2} + \frac{1}{\sigma_y^2} \right)^{-1} \\
m_{y,t} &= v_{y,t}\left( \frac{x_t w_x}{\sigma_x^2} + \frac{w_y m_{y,t-1} + w_{yz} m_{z,t}}{\sigma_y^2} \right) \\
v_{z,t} &= \left( \frac{w_{yz}^2}{\sigma_y^2} + \frac{1}{\sigma_z^2} \right)^{-1} \\
m_{z,t} &= v_{z,t}\left( \frac{(m_{y,t} - w_y m_{y,t-1})w_{yz}}{\sigma_y^2} + \frac{w_z m_{z,t-1}}{\sigma_z^2} \right).
\end{aligned}
$$

These computations can be done by first estimating $m_{y,t}, m_{z,t}$ before observing $x_t$ and then correcting using the observation. Let us show how it is done for $y$. We define $m'_{y,t}$ to be the posterior mean before observing $x_t$ (next step decoder prediction), $\tilde{m}_{y,t}$ (encoder posterior) to be the posterior means after observing $x_t$ but before updating the approximate posterior of higher latent variables ($z$ in this case), and finally $m_{y,t}$ to be the posterior mean after observing $x_t$ and updating all posteriors

(decoder, same step prediction). Thus, we can write

$$m'_{y,t} = w_y m_{y,t-1} + w_{yz} m'_{z,t} \tag{23}$$

$$\tilde{m}_{y,t} = v_{y,t} \left( \frac{x_t w_x}{\sigma_x^2} + \frac{m'_{y,t}}{\sigma_y^2} \right) = s_{y,t} \frac{x_t}{w_x} + (1 - s_{y,t}) m'_{y,t} \tag{24}$$

$$s_{y,t} = \text{sigmoid}(\log(\sigma_y^2 w_x^2 / \sigma_x^2)) \tag{25}$$

$$m_{y,t} = \tilde{m}_{y,t} + \frac{w_{yz} v_{y,t}}{\sigma_y^2} \left( m_{z,t} - m'_{z,t} \right), \tag{26}$$

which if we eliminate the tilded and primed $m$'s, give exactly the same equations as previously.

Generalizing the results for an arbitrary number of variables in the chain we have that for a generic level $l$ in a hierarchical chain, we have:

$$m'_{l,t} = w_l m_{l,t-1} + w_{l,l+1} m'_{l+1,t} \tag{27}$$

$$\tilde{m}_{l,t} = m'_{l,t} + \frac{(1 - s_l)}{w_{l-1,l}} (\tilde{m}_{l-1,t} - m'_{l-1,t}) \tag{28}$$

$$s_l = \frac{\frac{w_{l-1,l}^2}{\sigma_{l-1}^2}}{\frac{1}{\sigma_l^2} + \frac{w_{l-1,l}^2}{\sigma_{l-1}^2}} = \text{sigmoid}(\log(\sigma_l^2 w_{l-1,l}^2 / \sigma_{l-1}^2)) \tag{29}$$

$$m_{l,t} = \tilde{m}_{l,t} + s_l w_{l,l+1} (m_{l+1,t} - m'_{l+1,t}) \tag{30}$$

$$= \tilde{m}_{l,t} + s_l w_{l,l+1} (m_{l+1,t} - \frac{m'_{l,t} - w_l m_{l,t-1}}{w_{l,l+1}}) \tag{31}$$

These equations suggest a computational graph as shown in Fig. 9c. This is, if we literally interpret $\tilde{m}$ to be the activations of the encoder and $m$ and $m'$ to be the activations of the decoder, then we would conclude that we need upward-diagonal (the grey dotted arrow in Fig. 9c) decoder to encoder connections, and a recurrent decoder (curved dotted arrow in Fig. 9c). But we impose no such constraint, and the dotted lines are in fact only skip connections. So as long an information can be copied through across neural network layers (which it can), and as long as each layer has sufficient capacity, we can remove those skip connections and retain the same computational capabilities. This is what is done in RLadder. However, it would be interesting to see if adding those skip connections help with training on temporal tasks.

Using gatings in the bottom-up computations are justified by (28). Even though the top-down updates (27) and (31) are simple summations in the case of this linear Gaussian model, the more general gatings in the top-down pass provides more flexibility for nonlinear models. The activations in the level above can select one of a few possible evolution scenarios on the level below.

As we discussed previously, RLadder networks are usually trained by introducing tasks that encourage the network to learn the optimal inference of the latent variables. For temporal models, the natural low-level task is predicting the input at the next time instance. Thus the cost is $-\log(p(x_{t+1}|x_{1..t}))$:

$$C_{\text{pred}} = (\hat{x}_{t+1} - x_{t+1})^2$$

if we assume a Gaussian distribution for $\hat{x}$.

## D    Details of experiments

### D.1    Detailes of experiments with Occluded Moving MNIST

The RLadder architecture trained for the occluded moving MNIST dataset is presented in Table 5. The architecture of the baseline temporal model and the static feed-forward network trained on optimal digit reconstructions are shown in Tables 6–7. Initially, we trained the network only on the training set using the Adam optimizer [18] with an initial learning rate of 0.001. Every time the classification error on the validation set rose, we divided the learning rate by 2 until a minimum value of 0.0001. We then trained the network using both the training and validation sets following the learning rate schedule learned previously.

Table 5: Structure of RLadder for Occluded Moving MNIST

| | | Encoder | | Decoder | |
|---|---|---|---|---|---|
| Layer | Stride | # channels (size) | filter size | | filter size |
| Input | - | 1 | | | |
| ConvLSTM | 1 | 32 | (3, 3) | convG3 | (9,9) |
| Conv | 1 | 32 | (3, 3) | convG3 | (3, 3) |
| Conv | 1 | 32 | (3, 3) | convG3 | (3, 3) |
| Conv | 1 | 32 | (3, 3) | convG3 | (3, 3) |
| Pool | 2 | MAX | (2, 2) | convG3 | (6, 6) |
| ConvLSTM | 1 | 32 | (3, 3) | convG3 | (9, 9) |
| Conv | 1 | 64 | (3, 3) | convG3 | (3, 3) |
| Conv | 1 | 64 | (3, 3) | convG3 | (3, 3) |
| Conv | 1 | 64 | (3, 3) | convG3 | (3, 3) |
| Pool | 2 | MAX | (2, 2) | convG3 | (6, 6) |
| Conv | 1 | 128 | (3, 3) | convG3 | (3, 3) |
| Conv | 1 | 64 | (1, 1) | convG3 | (3, 3) |
| Conv | 1 | 32 | (1, 1) | convG3 | (3, 3) |
| Pool | 2 | AVG | (2, 2) | G1 | |
| LSTM | | 16 | | G1 | |
| SoftMax | | 10 | | - | |

Table 6: Temporal baseline network for Occluded Moving MNIST

| Layer | Stride | # channels (size) | filter size |
|---|---|---|---|
| Input | - | 1 | |
| Conv | 1 | 32 | (3, 3) |
| Conv | 1 | 32 | (3, 3) |
| Conv | 1 | 32 | (3, 3) |
| Pool | 2 | MAX | (2, 2) |
| Conv | 1 | 64 | (3, 3) |
| Conv | 1 | 64 | (3, 3) |
| Conv | 1 | 64 | (3, 3) |
| Pool | 2 | MAX | (2, 2) |
| Conv | 1 | 128 | (3, 3) |
| Conv | 1 | 64 | (1, 1) |
| Conv | 1 | 32 | (1, 1) |
| Pool | 2 | AVG | (2, 2) |
| LSTM | | 16 | |
| SoftMax | | 10 | |

Table 7: Feed-forward network in which the optimal temporal reconstruction of Occluded Moving MNIST is inputed for classification

| Layer | Stride | # channels (size) | filter size |
|---|---|---|---|
| Input | - | 1 | |
| Conv | 1 | 32 | (3, 3) |
| Conv | 1 | 32 | (3, 3) |
| Conv | 1 | 32 | (3, 3) |
| Pool | 2 | MAX | (2, 2) |
| Conv | 1 | 64 | (3, 3) |
| Conv | 1 | 64 | (3, 3) |
| Conv | 1 | 64 | (3, 3) |
| Pool | 2 | MAX | (2, 2) |
| Conv | 1 | 128 | (3, 3) |
| Conv | 1 | 64 | (1, 1) |
| Conv | 1 | 32 | (1, 1) |
| Pool | 2 | AVG | (2, 2) |
| SoftMax | | 10 | |

## D.2 Details of experiments with Polyphonic Music data

The RLadder architecture trained for the music dataset is presented in Table 8. We trained using the Adam optimizer with a learning rate of 0.01. Because the network overfitted very rapidly, we determined the optimal stopping time using the validation set and then trained on the training and validation set for the same number of epochs (between 3 and 10 depending on the hyperparameters).

Table 8: Structure of RLadder for Polyphonic Music dataset

| Layer | Stride | # channels (size) | filter size | | filter size |
|---|---|---|---|---|---|
| | | Encoder | | Decoder | |
| Input | - | 1 | | | |
| ConvLSTM | 1 | 32 | (1, 3) | ConvG2 | (1,3) |
| ConvLSTM | 2 | 64 | (1, 3) | ConvG2 | (1,3) |
| ConvLSTM | 2 | 96 | (1, 3) | ConvG2 | (1,3) |
| ConvLSTM | 2 | 128 | (1, 3) | ConvG2 | (1,3) |
| ConvLSTM | 2 | 160 | (1, 3) | ConvG2 | (1,3) |
| Pool | 2 | AVG | (1, 2) | G1 | |
| LSTM | | 96 | | G1 | |
| SoftMax | | 19 | | Identity | |

## D.3 Details of experiments with textured MNIST

The RTagger architecture used to train on Brodatz-textured MNIST data is in Table 9. The baseline convolutional network is model C from [24]. The networks were trained using the Adam optimizer with learning rate 0.0004. All training results were evaluated at update 160000. The size of the mini-batch was 16 for all the models. Batch normalization was applied to encoder and decoder layers.

Table 9: Structure of RTagger for textured MNIST dataset

| Layer | Stride | # channels (size) | filter size | Decoder |
|---|---|---|---|---|
| | | Encoder | | Decoder |
| Input | - | 1 | | |
| ConvLSTM | 2 | 96 | (8, 8) | ConvG1 |
| Conv | 1 | 96 | (5, 5) | ConvG1 |
| Conv | 1 | 96 | (5, 5) | ConvG1 |
| ConvLSTM | 2 | 192 | (4, 4) | ConvG1 |
| Conv | 1 | 192 | (5, 5) | ConvG1 |
| Conv | 1 | 192 | (5, 5) | ConvG1 |
| ConvLSTM | 2 | 192 | (4, 4) | ConvG1 |
| Conv | 1 | 192 | (5, 5) | ConvG1 |
| Conv | 1 | 192 | (5, 5) | ConvG1 |
| ConvLSTM | 2 | 11 | (4, 4) | ConvG1 |
| Pool | - | - | (6,6) | AVERAGE |
| FC | - | 11 | - | G1 |