[Reviews · NeurIPS 2017]

Reviewer 1



This paper presents a natural extension to ladder networks, which it interprets as implementing one step of message passing, to the recurrent case implementing multiple steps of massage passing amongst latent variables. The experiments seem to support this being a good move. I am fairly convinced of the value of the architecture being presented here, but the paper presents a few weaknesses which should be addressed. I will focus on the main two in this review. First, simply put: section 2 needs a formal description of the architecture being proposes, and it raises my eyebrows that the authors did not think this should be required in the body of the article. This can easily be rectified, and I am fairly confident that the description that will eventually be provided will match my educated guesses based on the diagrammatic representation of the model, but it’s poor form to not include it in the review draft. Second, the experimental section is adequate and convincing, but in 2017 (or in fact in 2015 or 2014) I roll my eyes at experiments involving MNIST. The structure of the occluded image experiment is cool, and I would love to see discussion of results on a similar set up with, say, CIFAR image classification. I don’t think this can be done on short notice for the response, but it would be nice to see in a final version of the paper, as the work would be stronger for it. Overall, not a bad paper, but could have been stronger. I think the idea is decent, so wouldn’t argue against it being accepted

Reviewer 2



In this paper, the authors introduce an extension of ladder networks for temporal data. Similarly to ladder network, the model proposed in the paper consists in an encoder-decoder network, with "lateral" connections between the corresponding layers of the encoder and the decoder. Contrary to the original model, recurrent ladder networks also have connection between different time steps (as well as skip connection from the encoder to the next time step). These recurrent connections allows to model temporal data or to perform iterative inference (where more than one step of message passing is required). The proposed model is then evaluated on different tasks. The first one is a classification task of partially occluded and moving MNIST digit, where the authors show that the proposed model performs slightly better than baselines corresponding to the proposed models with various ablation. The second task is generative modeling of various piano pieces. Finally, the last experiment is a classification task of MNIST digits with Brodatz textures, making the classification very challenging. While the ideas presented in this paper are natural and well founded, I believe that the current version of the papers has some issues. First, the proposed model is very succinctly described (in section 2), assuming that the reader is familier with ladder networks. This is not the case for me, and I had to look at previous papers to understand what is the model. I believe that the paper should be self-contained, which I do not think is the case for the submitted version. I also believe that some statement in the paper are a bit unclear or misleading: for example, I do not think that Fig. 1c corresponds to message passing with latent variables varying in time (I think that it corresponds to iterative inference). Indeed, in the case of a temporal latent variable model, there should also be messages from t to t-1. I also found the experimental section a bit weak: most of the experiments are performed on toy datasets (with the exception of the piano dataset, where the model is not state of the art). Overall, I am a bit ambivalent about the paper. While the ideas presented are interesting, I think that the proposed model is a bit incremental compared to previous work, and that the experiments are a bit weak. Since I am not familiar at all with this research area, I put a low confidence score for my review.

Reviewer 3



This paper presents a recurrent version of the Ladder network that was presented in NIPS 2015. The Ladder network, in brief, is a bottom-up/top-down hierarchical network with lateral connections between layers. The recurrent extension uses this network recurrently with connections between successive time slices. I am somehow divided about the value of this paper. From an architectural point of view, the contribution is minimal. It only consists of a recurrent connection that could be added to basically any network. Another contribution is the use of auxiliary loss functions. However, on this point, I am not clear whether the various losses are optimized jointly or alternately. More details would have been useful. However, the experimental results are interesting. A first experiment in Section 3.1 is with a Moving, Occluded version of the MNIST dataset prepared by the authors themselves. This experiment covers two reasonable baselines and various ablated versions. The best accuracy in Table 1 is achieved with all the proposed components. In addition, the next-frame predictions displayed in Fig. 4 illuminate about the internal behaviour. Also the semi-supervised experiment reported in Table 2 is interesting. The second experiment is a prediction of piano notes from music from 19 different classical composers. In this experiment, the proposed network performs well and is surpassed only by models that predict the simultaneous notes jointly. The experiment in section 4 shows that the proposed network not only can predict next-time frames, but next-time separate objects. This is done by using multiple such networks with tied parameters in parallel. Fig. 5 is another very interesting display of behaviour. The last experiment (Section 4.1) shows the remarkable performance of the network on a very challenging task where the MNIST digits have been almost completely destroyed by heavy texturing. Overall, in my opinion the experiments are very well-designed, diverse and probing and deserve attention for the proposed network.